# Dopamine neurons learn relative chosen value from probabilistic rewards

**Armin Lak\*†, William R Stauffer, Wolfram Schultz**

Department of Physiology, Development and Neuroscience, University of Cambridge, Cambridge, United Kingdom

**Abstract** Economic theories posit reward probability as one of the factors defining reward value. Individuals learn the value of cues that predict probabilistic rewards from experienced reward frequencies. Building on the notion that responses of dopamine neurons increase with reward probability and expected value, we asked how dopamine neurons in monkeys acquire this value signal that may represent an economic decision variable. We found in a Pavlovian learning task that reward probability-dependent value signals arose from experienced reward frequencies. We then assessed neuronal response acquisition during choices among probabilistic rewards. Here, dopamine responses became sensitive to the value of both chosen and unchosen options. Both experiments showed also the novelty responses of dopamine neurones that decreased as learning advanced. These results show that dopamine neurons acquire predictive value signals from the frequency of experienced rewards. This flexible and fast signal reflects a specific decision variable and could update neuronal decision mechanisms.

**\*For correspondence:** arminlak@gmail.com

**Present address:** †Institute of Ophthalmology, University College London, London, United Kingdom

## Introduction

Individuals frequently make predictions about the value of future rewards and update these predictions by comparison with experienced outcomes. A fundamental determinant of reward value is reward probability (*Pascal, 2005*). When an environmental cue predicts reward in a probabilistic fashion, the brain needs to learn the value of such a cue from the frequency of experienced rewards. Such learning enables individuals to compute the economic value of environmental cues and thus allows forpa efficient decision making.

The phasic activity of dopamine neurons encodes reward prediction error (*Schultz et al., 1997*; *Bayer and Glimcher, 2005*; *Enomoto et al., 2011*; *Cohen et al., 2012*). These prediction error responses increase monotonically with the expected value of reward, including reward probability (*Fiorillo et al., 2003*; *Tobler et al., 2005*). Cues that predict reward with high probability evoke larger responses than cues predicting the same reward with lower probability (*Fiorillo et al., 2003*). Moreover, during an economic choice task, responses of dopamine neurons and striatal dopamine concentration reflect the reward probability of the cue the animal has chosen (*Morris et al., 2006*; *Saddoris et al., 2015*). In these studies, neuronal responses to reward predicting cues were examined only after the animals received substantial training with the same reward-predicting cues.

The responses of dopamine neurons have been also examined during learning (*Mirenowicz and Schultz, 1994*; *Hollerman and Schultz, 1998*). These studies primarily focused on how dopamine responses to rewards develop during learning of cue-reward association. This neuronal acquisition happens gradually (*Hollerman and Schultz, 1998*), and is well-approximated by reinforcement learning (RL) models (*Pan et al., 2005*). Similarly, striatal dopamine concentration reflects values of probabilistically delivered rewards during learning (*Hamid et al., 2016*). However, it remains unknown how learning about probabilistic rewards shapes responses of dopamine neurons to reward predicting cues, and how this neuronal learning participates in decision making.

**eLife digest** Learning to choose the most fulfilling reward from a number of options requires the ability to infer the value of each option from unpredictable and changing environments. Neurons in the brain that produce a chemical called dopamine are critical for this learning process. They broadcast a 'prediction error' signal that alerts other areas of the brain to the difference between the actual reward and the previously predicted reward. Previous studies have shown that when dopamine neurons signal a prediction error, new learning about the value of an option takes place.

To understand exactly what happens during this learning process, Lak et al. recorded electrical activity from dopamine neurons in the brains of two monkeys. Over a number of trials, the monkeys were shown one of three different novel images, each of which was associated with a different likelihood of receiving a large amount of a fruit juice reward.

The recordings showed that the dopamine response to cues was divided into early and late components. At the start of learning, when the monkeys were unfamiliar with the likelihood that each image would yield a large juice reward, the early part of the dopamine response was large. The size of this part of the response decreased as the monkeys became more familiar with each image. The later part of the dopamine response changed to reflect the rewards the monkeys had received on previous trials. On trials where a reward was delivered, this part of the response grew larger, but diminished if a reward was not given.

When the monkeys had to choose between rewards, the dopamine response was larger when the monkey chose the higher valued option over the lesser valued one, and smaller when the opposite choice was made, thus reflecting the animal's choice. These choice-dependent responses were also sensitive to the value of unchosen option, and therefore, reflected the difference between the value of chosen and unchosen options.

Future studies are now required to find out how manipulating the activity of the dopamine neurons influences the way animals learn and make decisions.

We addressed these questions by recording the activity of dopamine neurons in monkeys during the learning of novel cues predicting specific reward probabilities. We studied dopamine responses during both simple Pavlovian conditioning and during risky choices. In both tasks, dopamine responses to cues showed two distinct response components: an early component reflecting novelty, and a later component that developed during learning to encode the value of probabilistic rewards acquired from experienced reward frequencies. Reinforcement learning models served to separate these two components more formally. During choice, the acquired dopamine responses reflected the value of the chosen option relative to the unchosen option.

## Results

### Pavlovian learning of probabilistic rewards

Pavlovian conditioning is the most basic mechanism by which an organism can learn to predict rewards. This behavioural paradigm provides a straightforward platform for monitoring neuronal correlates of learning. To investigate how responses of dopamine neurons develop during learning to reflect the value of probabilistically delivered rewards, we monitored two monkeys during a Pavlovian conditioning task (*Figure 1A*). Visual cues (fractal pictures, never seen before) predicted gambles between a large (0.4 ml) and a small (0.1 ml) juice reward, delivered 2 s after cue onset. The probability of receiving the large reward was $p=0.25$, $p=0.50$, or $p=0.75$; the probability of small reward was correspondingly $1 - p$. In each learning block, three novel cues were differentially associated with three different reward probabilities, but only one cue was presented to the animal on each trial. This situation conforms to a learning set in which the animals learned to rapidly assign in each learning block one of the three possible probabilities to each cue (*Harlow, 1949*).

The animals gradually developed differential anticipatory lick responses, measured between cue onset and reward delivery, over about 10 trials for each cue (*Figure 1B*, $p<0.01$ in both animals,

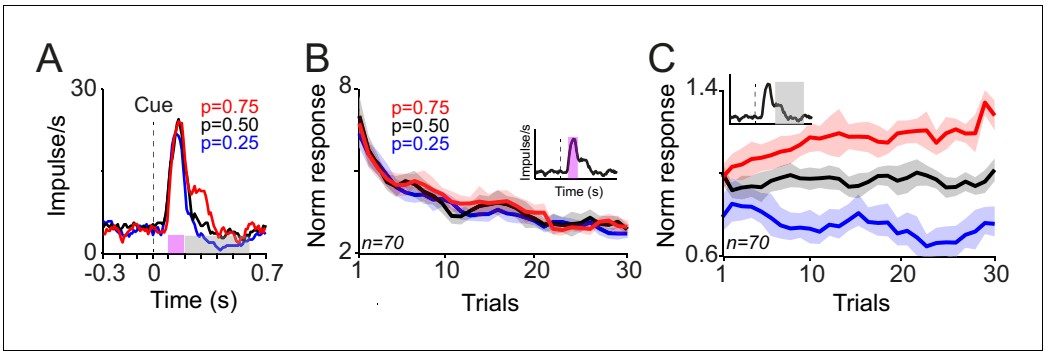

**Figure 1.** Monkeys rapidly learn the value of cues that predict rewards with different probabilities. (**A**) Pavlovian task. Left: example of novel visual cues (fractal images) presented to monkeys. In each trial, animals were presented with a visual cue and received a large (0.4 ml) or small (0.1 ml) drop of juice reward 2s after cue onset. Specific cues predicted the large reward with probabilities of p=0.25, p=0.5 and p=0.75, together with small reward at 1–p. In each session of the experiment (lasting 90–120 trials), three novel cues were differentially associated with the three tested reward probabilities. Over consecutive trials, cues with different reward probabilities were presented to animals pseudorandomly. Trials were separated by inter-trial intervals of 2–5 s. Animals had no specific behavioural requirements throughout this task. (**B**) Monkeys' lick responses during Pavlovian learning. The lick responses were measured from cue onset to onset of reward delivery.

one-way ANOVA). This result suggests that the monkeys learned the value of novel sensory cues that predicted rewards with different probabilities.

We recorded the responses of 38 and 32 dopamine neurons in monkeys A and B, respectively, during this Pavlovian learning task. The neuronal responses to cues showed two components (*Figure 2A*), analogous to previous studies (*Nomoto et al., 2010*; *Stauffer et al., 2014*; *Schultz, 2016*). Specifically, an early activation at 0.1–0.2 s after cue onset most likely reflected the previously observed novelty signals (*Ljungberg et al., 1992*; *Horvitz et al., 1997*; *Schultz, 1998*; *Costa et al., 2014*; *Gunaydin et al., 2014*). It decreased progressively during learning blocks (*Figure 2B*, *Figure 2—figure supplement 1A*; and 55/70 neurons; p<0.05 power function fit to trial-

**Figure 2.** Responses of dopamine neurons acquire predictive value from the frequency of rewards. (**A**) Peri-stimulus time histograms (PSTHs) of a dopamine neuron in response to novel cues predicting rewards with different probabilities. Pink (0.1–0.2 s after cue onset) and grey (0.2–0.6 s after cue onset) horizontal bars indicate analysis windows used in B and C, respectively. (**B**) Decrease of neuronal population responses, measured at 0.1–0.2 s after cue onset (pink inset), over consecutive learning trials. Error bars show standard error of mean (s.e.m.) across neurons (n = 70, pooled from monkeys A and B). (**C**) Differentiation of neuronal population responses, measured at 0.2–0.6 s after cue onset (grey inset), over consecutive learning trials. The following figure supplement is available for *Figure 2*:
The following figure supplement is available for figure 2:

**Figure supplement 1.** Compound novelty-value responses of dopamine neurons to novel cues associated with different probabilistic rewards.

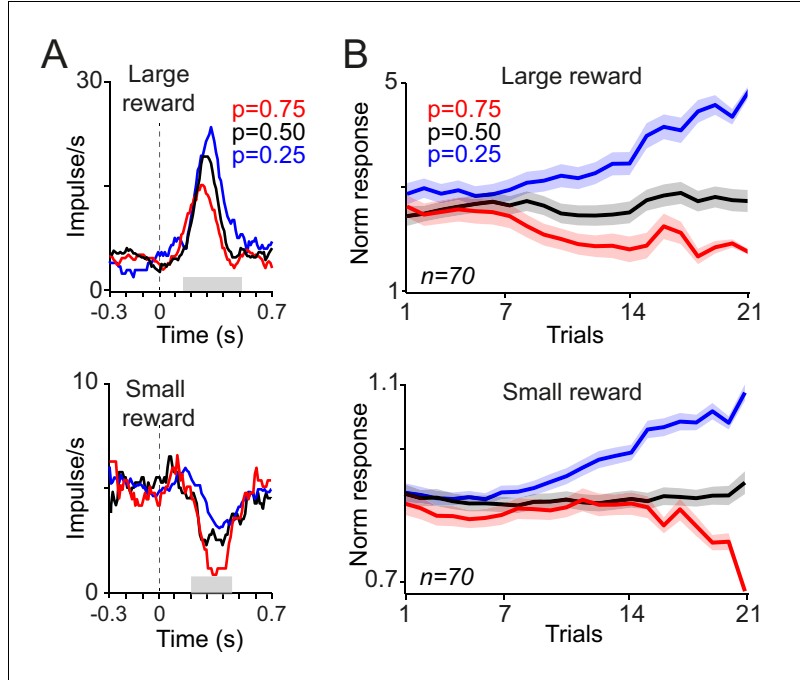

**Figure 3.** Responses of dopamine neurons to reward delivery develop over trials to reflect the learned value of probabilistic cues. (**A**) PSTHs of example dopamine neurons in response to delivery of large and small juice rewards (top, bottom). Probabilities indicated in colour refer to the occurrence of the large reward in gambles containing one large and one small reward (0.4 ml and 0.1 ml, respectively). (**B**) Neuronal population responses to large and small juice rewards over consecutive learning trials. Responses were measured in analysis windows indicated by corresponding grey horizontal bars in A (top: 0.15–0.5 s, bottom: 0.2–0.45 s after reward onset).

by-trial responses), reflecting cue repetition better than number of consecutive trials ($R^2 = 0.86$ vs. $R^2 = 0.53$, linear regression). This response component failed to differentiate between the cues predicting different reward probabilities (*Figure 2B*, p=0.61, one-way ANOVA).

In contrast to the initial novelty response, a subsequent response component occurred at 0.2–0.6 s after cue onset and became differential during the learning of different reward probabilities (*Figure 2C*, 26/70 neurons, one-way ANOVA on responses from sixth to last trial, p<0.05). These responses became statistically distinct after experiencing each cue six times (*Figure 2C*, p<0.01, one-way ANOVA on trial-by-trial neuronal population responses). Thus, throughout each short learning session with a new set of fractal images, a considerable fraction of dopamine neurons learned the value of reward predicting cues from the frequency of experienced rewards. Analysis on the whole duration of neuronal response (0.1–0.6 s after cue onset) showed that the compound novelty-value responses decreased over consecutive learning trials and also reflected the learned value of cues (*Figure 2—figure supplement 1B*). Taken together, these results demonstrate how dopamine neurons gradually acquire probability-dependent value responses from the frequency of experienced rewards, and how these responses differ from their novelty responses.

Examination of dopamine prediction error responses to reward delivery provided further evidence for neuronal acquisition of reward probability. Neuronal responses to reward developed gradually to reflect the values of the cues. Specifically, activating neuronal responses to large reward (0.4 ml) were larger after cues that predicted this outcome with lower probability, compared to cues predicting the same outcome with higher probability (*Figure 3A* top). Conversely, depressant neuronal responses to small reward (0.1 ml) were more pronounced after cues predicting large reward with higher probability (*Figure 3A* bottom). Thus, both the activating and depressant responses were consistent with reward prediction error coding. The neuronal responses to both large and small rewards differentiated gradually over consecutive trials, based on the predicted probability of getting each of those rewards, and reached statistical significance after 9 and 16 trials, respectively

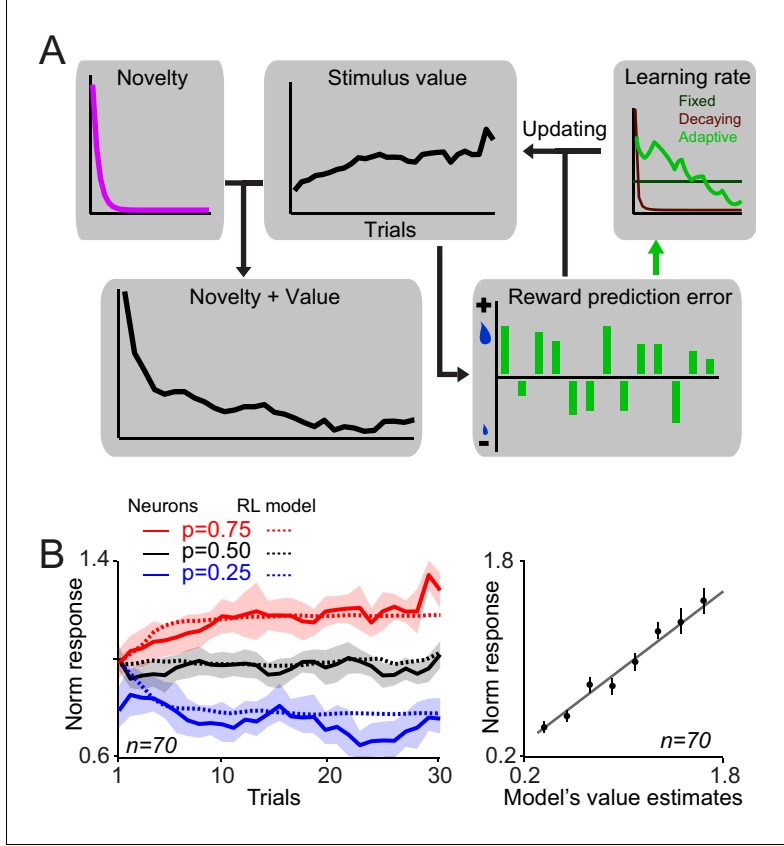

**Figure 4.** A reinforcementlearning model with a novelty term and an adaptive learning rate account for dopamine responses during learning. (**A**) Schematic of RL models fitted on neuronal responses. In each trial, the model updates the value of stimulus based on the experienced reward prediction error. Six variants of RL models were tested (three different learning rates, each with or without novelty term). In brief, we optimized the free parameters of each model so that it minimized the difference between dopamine responses to cues (measured 0.1–0.6 s after the cue, thus including both novelty and value component) and model's estimates of novelty + value. We then examined the relation between value-driven neuronal responses and value estimates of the superior model and also the relation between novelty-driven neuronal responses and novelty estimates of the superior model. For details of model implementation and fitting procedure see Materials and methods. (**B**) Left: Value estimates of the superior model (i.e. the model with a novelty term and adaptive learning rate) overlaid on neuronal population responses measured 0.2–0.6s after the cue onset,(from *Figure 2C*). For details of parameter estimation and model comparison see *Supplementary file 1.* Right: Regression of dopamine responses to cues (dopamine value responses, i.e. 0.2–0.6 s after the cue onset) onto value estimates of the superior RL model. See *Figure 4—figure supplement 1* for regression of dopamine novelty signals onto novelty-driven model's estimates.

The following figure supplement is available for figure 4:

**Figure supplement 1.** A reinforcement learning model with a novelty term and an adaptive learning rate account for dopamine responses during learning.

(*Figure 3B*, p<0.02, one-way ANOVA on neuronal population responses). The development of dopamine responses to rewards further suggests that early and late responses to cues convey distinct signals. If early responses to cues contained predictive values signals (i.e. reflecting an optimistic value initialisation), such signals should have contributed to prediction error computations at reward time. However, the pattern of neuronal reward prediction errors (*Figure 3B*) suggests that these responses were computed in relation to late responses to cues, and reflected cue values initialised around the average value of all cues. Accordingly, neuronal responses to rewards were accounted for by the late component of neuronal responses to cues as well as the received reward size, with no significant contribution from the early component of cue responses (p=0.0001, 0.43

and 0.021 for reward size, early and late cue responses, respectively; multiple linear regression). Thus, the development of the prediction error responses at the time of reward reflect the acquisition of probability-dependent value responses to cues; dopamine neurons learn the value of novel cues and use these learned values to compute prediction errors at the time of the outcome.

Dopamine responses to rewards and reward-predicting cues are described well by prediction errors derived from standard reinforcement learning (RL) models. These models calculate trial-by-trial prediction errors and use these values, weighted by a learning rate parameter, to update associative strengths. While it is straightforward to see that in the RL framework positive and negative reward prediction errors, encountered upon receiving large and small rewards, can lead to reward probability-dependent cue responses, it is not clear what form of RL model can best account for the development of value and novelty driven dopamine responses during learning. We therefore investigated different variants of RL models to discover which RL variant can best capture the observed development of dopamine responses.

We devised models with three different types of learning rates: (1) a learning rate which was fixed over trials resembling the original Rescorla-Wagner model (*Rescorla and Wagner, 1972*), (2) a learning rate that decayed over trials thus representing the idea that updating should occur faster in early trials, and (3) a learning rate that was adaptively adjusted on every trial based on past prediction errors, thus capturing the idea that learning is slower when prediction errors are negligible (*Pearce and Hall, 1980*; *Pearce et al., 1981*; *Le Pelley, 2004*). For each of the three learning rate-type models, we fit the data with and without the presence of a term for the novelty which decayed over trials (*Figure 4A*, see Materials and methods) (*Kakade and Dayan, 2002*). Thus, in total we explored six model variants, and fit the models to the dopamine responses using the rewards actually delivered during the experiments (Materials and methods).

The model that included an adaptive learning rate and novelty term outperformed all other model variants in accounting for dopamine responses (*Figure 4B* left and *Figure 4—figure supplement 1A*, see *Supplementary file 1* for details of parameter estimation and model comparisons). Consistent with this, regression of second component dopamine responses to cues (0.2–0.6 s after the cue onset) onto value estimates of the superior model was highly significant (*Figure 4B*, right, $R^2$ = 0.93, p=0.00003). Moreover, regression of dopamine novelty responses to model-driven novelty estimates was statistically significant (*Figure 4—figure supplement 1B*, $R^2$ = 0.63, p=0.001). In this simulation, the estimated learning rate decayed over trials, while fluctuating based on past prediction errors (*Figure 4A* and *Figure 4—figure supplement 1C*, *Supplementary file 1*). The model fittings suggested that the development of early and late responses to cues follows different temporal dynamics. In a model variant that is rearranged to include the novelty decay term as an error-driven learning process (simulating optimistic value initialisation), the recovered learning constant for the first component of cue responses was significantly larger than the recovered learning constant for the late response component (*Figure 4—figure supplement 1D*, p<0.0001, Mann-Whitney U test, see Materials and methods). This observation suggests that early and late components of dopamine responses follow distinct temporal dynamics. Together, these results suggest that, during learning about probabilistic rewards, the trial-by-trial dopamine responses to cues adjust according to how much learning has advanced. Neuronal responses to cues rapidly develop early during learning and value updating becomes slower as learning progresses and prediction errors become smaller.

## Learning the value of probabilistic rewards during choice

Our findings so far demonstrate that dopamine neurons acquire the value of probabilistic rewards during a Pavlovian learning paradigm. We next investigated the behavioural and neuronal signatures of learning about probabilistic rewards during a decision task. As before, we used new fractal stimuli for each learning episode, which also prevented a carry-over of learned pictures from the Pavlovian to the choice task.

The animals made saccade-guided binary choices between two cues (*Figure 5A*). The animals had extensive prior experience with one of the cues (familiar cue) that predicted a 50% chance of receiving a large reward (0.4 ml) and 50% chance of receiving a small reward (0.1 ml). In each block (typically 50 trials) the familiar cue was always offered as one choice alternative. The other cue was novel, and its reward probability was unknown to the animal. Similar to the Pavlovian task, the novel cues were associated with reward probabilities of 0.25, 0.50 or 0.75 of receiving the large (0.4 ml) reward and 0.1 ml otherwise. This situation resembled a learning set (*Harlow, 1949*) in which the

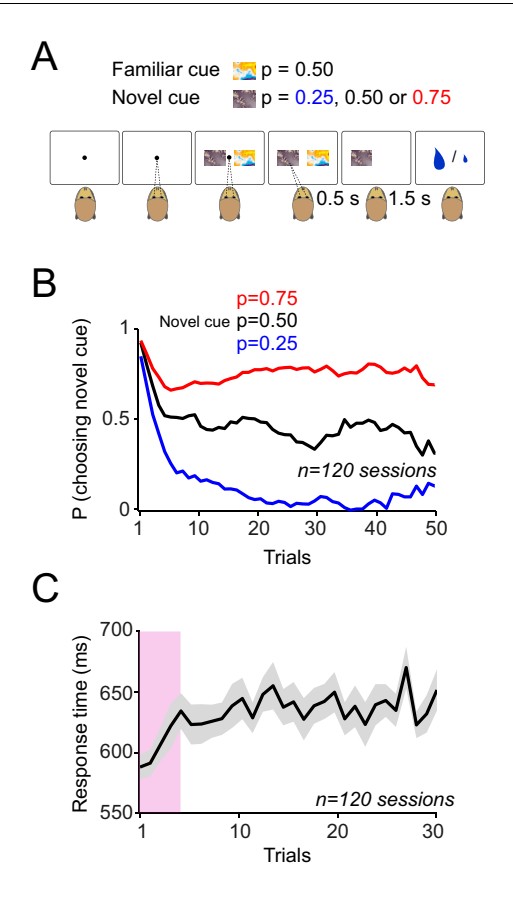

**Figure 5.** Monkeys rapidly learn to make meaningful choices among probabilistic reward predicting cues. (**A**) Choice task. In each trial, after successful central fixation for 0.5 s, the animal was offered a choice between two cues, the familiar cue and the novel cue. The animal indicated its choice by a saccade towards one of the cues. The animal was allowed to saccade as soon as it wanted. The animal had to keep its gaze on the chosen cue for 0.5 s to confirm its choice. Reward was delivered 1.5 s after the choice confirmation. The animals had extensive prior experience with one of the cues (familiar cue predicting 50% chance of getting 0.4 ml and 50% chance of receiving 0.1 ml). The alternative cue was a novel cue with the reward probability unknown to the animal. The novel cues were associated with reward probabilities of 0.25, 0.50 or 0.75 of receiving the large (0.4 ml) reward and 0.1 ml otherwise. After a block (of typically 50 trials) the novel cue was replaced with another novel cue. Trials were separated with inter-trial interval of 2–5 s. Failure to maintain the central fixation or early breaking of fixation on the chosen option resulted in 6 s time-out. (**B**) Monkeys' choice behaviour. At the onset of each learning session, both animals chose the novel cue over the familiar cue for 4–5 trials. Afterwards, animals preferentially chose the cue that predicted reward with higher probability. (**C**) Saccadic choice response times. Both monkeys showed significantly faster reaction times (defined as the interval between the cue onset and the time the animal's saccade acquired the chosen option) in the first 4–5 trials of each learning block. Error bars are s.e.m across behavioural sessions.

animals rapidly learned to assign one of three possible values to the novel cue. In this task monkeys had to choose the novel cue in order to learn its reward probability. At the onset of each learning block, both monkeys consistently selected the novel cue in the first few trials (*Figure 5B*, p<0.01 in both animals, Mann-Whitney U test on choice probabilities in trials 1–4 versus later trials). This exploratory behaviour was accompanied by shorter saccadic response times (measured between cue onset and saccadic acquisition of the chosen option), compared to the response times observed during later trials when the highest probability option was usually chosen (*Figure 5C*, p<0.01 in both animals, Mann-Whitney U test on trials 1–4 versus later trials). After five trials, both animals chose the higher option 75% of the time (p<0.001; one-way ANOVA on choice probabilities). These results

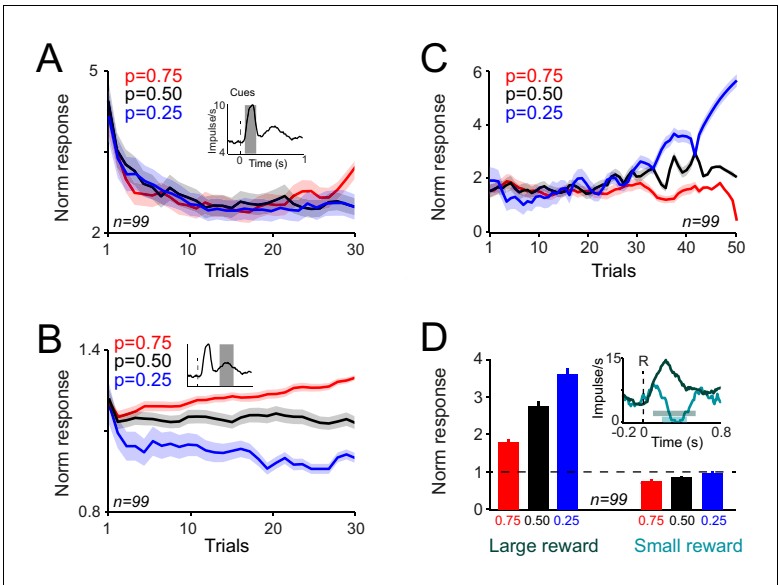

**Figure 6.** Dopamine responses to cues differentiate as monkeys learn the value of novel cues in the choice task. (**A**) Neuronal population responses to cues over consecutive trials of the choice task, measured during 0.1–0.2 s after the cue onset (Dopamine novelty responses, see inset). Only trials in which animal chose the novel cue were shown in all panels of this figure. (**B**) Neuronal population responses to cues over consecutive trials of the choice task, measured during 0.4–0.65 s after the cue onset (Dopamine value responses, see inset). See *Figure 6—figure supplement 1* for more detailed analysis of time course of the neuronal activity. (**C**) Population dopamine responses to the large reward over trials in which the novel cue was chosen and large reward was delivered. (**D**) Population dopamine responses to the reward delivery in trials in which the novel cue was chosen. Each bar demonstrates the mean neuronal response averaged across later (30th to last trial) of each session. Bars on the left represent neuronal activity in response the large reward (0.4 ml). Bars on the right represent neuronal activity in response to the small reward (0.1 ml). Inset illustrates PSTHs of an example neuron in response to small and large rewards. Horizontal bars in the inset indicate the temporal window used for computing bar plots (large rewards: 0.1–0.55 s after the reward onset, small rewards: 0.2–0.45 s after the reward onset). Error bars represent s.e.m across neurons (n = 99, pooled from monkeys A and B).

The following figure supplement is available for figure 6:

**Figure supplement 1.** Neuronal responses to cue in the choice task.

suggest that the animals rapidly learned the value of novel reward predicting cues in the choice task and used these learned values to make efficient economic choices.

We recorded dopamine activity during the choice task (57 and 42 neurons in monkey A and B). In order to examine neuronal signatures of probability-dependent value learning, we first focused on trials in which animal choose the novel cues. Neuronal responses immediately after the cue onset (0.1–0.2 s after the cue onset) decreased over consecutive trials, reflecting the stimulus novelty (76/99 neurons, power function fit on trial-by-trial responses, $p < 0.05$), but never differentiated to reflect the learned value of cues (*Figure 6A*, $p > 0.1$ in both animals, one-way ANOVA on population responses). In contrast, the later component of the neuronal response (0.4 to 0.65 s after the cue onset) developed differential responses that reflected the learned value of cues (*Figure 6B*, $p < 0.01$ from fifth trial onwards, one-way ANOVA on neuronal population responses). Neuronal activity between these two windows of analysis reflected a smooth transition from encoding stimulus novelty to encoding the learned value signals (*Figure 6—figure supplement 1*). These results indicate that during economic choices, dopamine responses contain two distinct components; the first component of the neuronal responses mainly reflects the stimulus novelty, whereas the second component of neuronal activity differentiates during learning to encode the learned value of cues.

We then explored signatures of value learning in neuronal responses to rewards. We analysed the dopamine responses to rewards, focusing again on trials in which animals chose the novel cue.

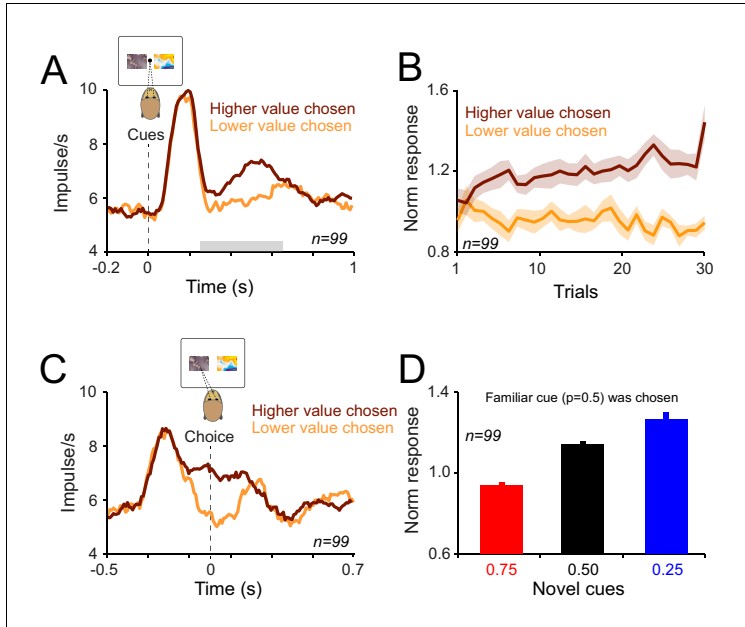

**Figure 7.** During learning dopamine neurons acquire choice-sensitive responses which emerge prior to response initiation. (**A**) Population dopamine PSTHs to cues in the choice task. Grey horizontal bar indicates the temporal window used for statistical analysis. In all plots, all trials of learning blocks are included. Note that the results would be similar after excluding initial trials of each learning session. (**B**) Population dopamine responses to cues (0.4–0.65 s after the cue onset) over consecutive choice trials. Trials are separated based on animal's choice. (**C**) Population dopamine PSTHs aligned to the saccade initiation (i.e. the time on which animal terminated the central fixation to make a saccade towards one of the cues). Dopamine choice-sensitive responses appeared ~130 ms prior to saccade initiation. (**D**) Averaged neuronal population responses to cues in trials in which animals chose the familiar cue. Despite the fact that animal had extensive experience with the familiar cue (and hence accurate estimate of its value), neuronal responses showed dependency on the value of the unchosen cue. See **Figure 7—figure supplement 1** for the time course of this effect over consecutive trials of learning.

The following figure supplement is available for figure 7:

**Figure supplement 1.** Population dopamine responses to cues over trials in which animals chose the familiar cue over the novel cues.

Following the initial trials of each learning block, neuronal responses to reward began to reflect the probability predicted by the chosen cue (**Figure 6C**, p<0.01 after 36 choice trials; one-way ANOVA on responses to the large reward). In accordance with reward prediction error coding, the responses to 0.4 ml juice were significantly larger when the chosen cue predicted this outcome with lower compared to higher probability (**Figure 6D** left, p<0.001 in both animals, one-way ANOVA on neuronal responses averaged from 30th to the last trial of each block). Similarly, the negative prediction error responses to the small (0.1 ml) rewards were more pronounced (i.e. stronger depression of activity) when the chosen cue predicted this outcome with lower probability (**Figure 6D** right, p=0.02, one-way ANOVA on neuronal responses averaged from 30th to last trial of the block). Together, these results indicate that during a learning task that included economic choice, dopamine neurons learn the value of novel cues from the probabilistic outcomes associated with those cues and compute reward prediction errors by comparing these learned values with the actual trial outcome.

## Choice-dependent dopamine responses

To investigate whether dopamine responses depended on the animals' choice, we divided the trials according to the choice that the animals made (i.e. lower probability chosen or higher probability chosen), and examined the trial-by-trial neuronal responses to cue presentations. The magnitude of the neuronal response depended on the choice (**Figure 7A**). Larger neuronal responses occurred

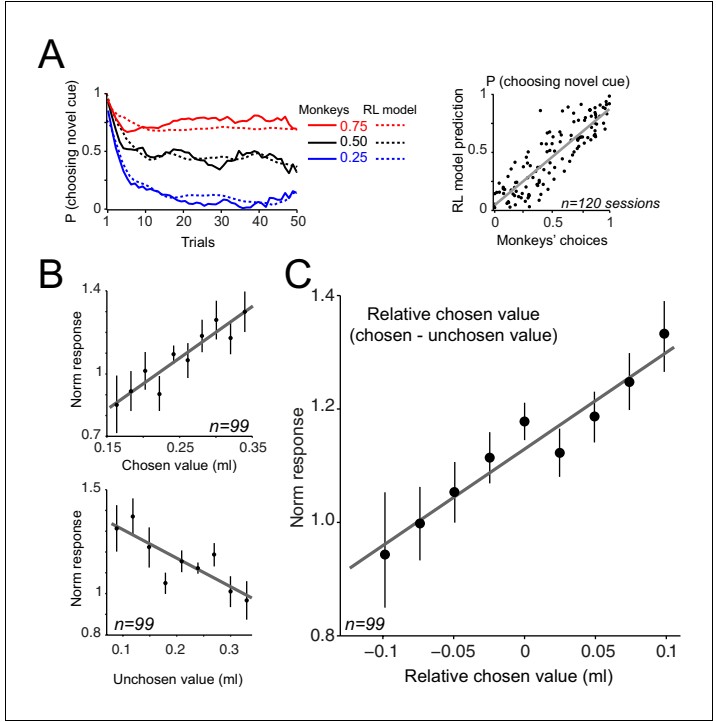

**Figure 8.** Dopamine neurons encode relative chosen values. (**A**) Left: Animals choices were simulated using standard reinforcement learning (RL) models (see *Figure 8—figure supplements 1* and *2* and Materials and methods). Dotted lines show the performance of the model in predicting monkeys' choices. Solid lines show monkeys' choice behaviour (identical to *Figure 5B*). The parameters of the RL model were separately optimized for each behavioural session (*Supplementary file 2*). Right: The RL model's session-by-session probability of choosing the novel cue, estimated using model's optimized parameters, versus monkeys' session-by-session probability of choosing the novel cue. (**B**) Upper panel: Regression of neuronal population responses to cues onto trial-by-trial chosen values estimated from the RL model fitted on monkeys' choice data. Lower panel: Regression of neuronal population responses to cues onto trial-by-trial unchosen values estimated from the RL model fitted on the choice data. (**C**) Regression of neuronal population responses to cues onto trial-by-trial relative chosen values (i.e. chosen value – unchosen value) estimated from the RL model fitted on the choice data. Importantly, the chosen and unchosen value variables were not, on average, strongly correlated (r = −0.039, Pearson's correlation), and we excluded from this analysis sessions in which the absolute value of the correlation coefficient between the chosen and unchosen variables was larger than 0.25. In B and C, the neuronal responses were measured 0.4–0.65 s after cue onset (i.e. dopamine value signals) and are regressed against value estimates of the superior model. In explaining the neuronal responses, relative chosen value outperformed other variables in all six models tested. See *Figure 8—figure supplement 2B* for regression of responses measured 0.1–0.2 s after cue onset (i.e dopamine novelty responses) onto model-driven novelty estimates. Regression of whole neuronal responses (0.1–0.65 s after the cue onset) against value estimates of the RL model further confirmed relative chosen value as the best explanatory variables ($R^2$ = 0.57, 0.61 and 0.83 for unchosen, chosen and relative chosen values). In all plots, all trials of learning blocks are included (regression results are similar after excluding initial (i.e. 5) trials of each session).

The following figure supplements are available for figure 8:

**Figure supplement 1.** Schematic of the RL model used for simulating monkeys' choice behaviour.

**Figure supplement 2.** Estimated learning rates of the RL model and regression of dopamine novelty responses to model-driven novelty estimates.

when the animal chose the higher probability (more valuable) option, compared to the lower probability (less valuable) option (*Figure 7A*, p<0.02, Mann-Whitney U test on population responses during 0.25–0.65 s after cue onset in both animals and p<0.02 in 11 out of 99 single cells, Mann-

Whitney U test). The early response component (0.1–0.2 s after cue onset) did not reflect animals' choice (p>0.1 in both animals, Mann-Whitney U test). The choice-sensitivity of neuronal responses developed rapidly during learning; they reached statistical significance after five choice trials (*Figure 7B*, p<0.01 from fifth trial onwards, one-way ANOVA). Within a given trial, choice predictive activity arose as early as 130 ms prior to saccade onset (*Figure 7C*, analysis window starting 0.2 s before the choice onset, p<0.01 from 130 ms before saccade onset, Mann-Whitney U test). These results demonstrate that during learning dopamine responses rapidly develop choice sensitivity and reflect the value of the option chosen by the animal (i.e. chosen value). Furthermore, these neurons began encoding this decision variable even before the overt choice (i.e. onset of saccade) occurred.

We next investigated whether neuronal responses during choice could also reflect the value of unchosen option. We inspected trials in which the animals chose the familiar cue, and divided those trials according to the reward probability of the novel cue. Despite the fact that the animals had extensive experience with the familiar cues (and hence accurate estimates of their value), the neuronal responses during choices of the familiar cue were significantly larger when the alternative option predicted low compared to high probability of the same large reward (*Figure 7D*, p=0.01, one-way ANOVA, see *Figure 7—figure supplement 1* for the time course of the effect). Together, these results suggest that during choices among probabilistic rewards, dopamine responses are sensitive to the value of both chosen and unchosen options.

We used RL models, similar to those described in the Pavlovian experiment but modified to account for choice, to explore the observed neuronal coding in a trial-by-trial fashion (*Figure 8—figure supplement 1*, see Materials and methods). The model with adaptive learning rate and novelty term outperformed all other models in accounting for animals' choices (see *Supplementary file 2* for parameter estimation and model comparison). This model accounted well for animals' trial-by-trial choices throughout learning blocks (*Figure 8A*, left) as well as their session-by-session preferences (*Figure 8A*, right, r = 0.86, p<0.0001). Similar to the Pavlovian experiments, the estimated learning rate exhibited a decay over trials while maintaining sensitivity to past prediction errors (*Figure 8—figure supplement 2A*). In all tested models, the estimated learning of the novel cue was larger than the estimated learning rate of the familiar cue, indicating that during learning animals updated the value of novel cue more than the value of familiar cue (*Supplementary file 2*). Linear regression of the neuronal responses (measured 0.4–0.65 s after the cue onset) onto model's value estimates revealed a positive relationship to chosen values and an negative relationship to unchosen value (*Figure 8B*, chosen value: $R^2 = 0.65$, p=0.005, unchosen value: $R^2 = 0.84$, p=0.0001, single linear regression). However, a relative chosen value variable, defined as chosen value – unchosen value, fit the data far better, compared to the chosen or unchosen value variables (*Figure 8C*, $R^2 = 0.91$, p=0.00005, single linear regression, p<0.02 in 15 out of 99 single cells), confirming earlier results shown in *Figure 7*. Similar to the Pavlovian experiment, regression of dopamine novelty responses (0.1–0.2 s after the cue) onto model's novelty estimates was significant ($R^2 = 0.61$, p=0.001, *Figure 8—figure supplement 2B*). Together, these results suggest that when the animals learn to choose among probabilistic rewards, dopamine neurons took the value of both chosen and unchosen options into account and thus reflected relative chosen value.

## Discussion

Building on previous findings that the prediction error responses of dopamine neurons increase monotonically with reward probability and expected value (*Fiorillo et al., 2003*; *Tobler et al., 2005*), this study shows how these probability dependent value responses evolve through learning. Dopamine responses showed two distinct response components. Responses immediately after the cues decreased as learning advanced, reflecting novelty. The second response component developed during learning to encode the value of probabilistic rewards acquired from experienced reward frequencies. Correspondingly, the prediction error responses at reward time changed over the course of learning to gradually reflect the learned reward values. Results from previous studies on fully established tasks suggest that the acquired dopamine responses to probabilistic rewards do not code reward probability on its own but rather increase monotonically with the statistical expected value (*Fiorillo et al., 2003*; *Tobler et al., 2005*). The present learning data are fully compatible with those results. During choices, the acquired dopamine value signals coded the value of the chosen option relative to the unchosen option. These results are consistent with previous

findings that showed chosen value coding in dopamine neurons (*Morris et al., 2006*). However, we provide new evidence in favour of a more nuanced, relative chosen value coding scheme whereby dopamine responses also reflect the value of un-chosen option. Together, our data suggest that dopamine neurons extract predictive reward value from the experienced reward frequency and code this information as relative chosen value.

Throughout learning, dopamine responses to cues developed to reflect the value of upcoming rewards, indicating that these neurons extract predictive value signals from experienced reward frequencies. In the learning experiment that involved choices, the neuronal responses rapidly differentiated to reflect animal's choice. These differential responses, despite appearing more than 100 ms prior to overt behaviour, reflect prediction errors in relation to an already computed choice, and thus might not directly participate in current choice computation. Our modelling results provided further insights into the dynamics of neuronal learning process. First, the development of neuronal responses over trials as well as animals' choices were best explained by models that adaptively adjusted their learning rate based on past prediction errors, resembling previous studies in human subjects (*Nassar et al., 2010*; *Diederen and Schultz, 2015*). Second, value-dependent dopamine responses were still updated even after the dopamine novelty responses stabilized, suggesting two distinct time courses for these two components of neuronal activity. Interestingly, in both Pavlovian and choice tasks, behavioural preferences as well as neuronal responses to cues reflected reward probability earlier during learning than the neuronal reward responses. This temporal difference might suggest an origin of behavioural preferences and acquired dopamine cue responses in other brain structures, rather than relying primarily on dopamine reward prediction error signals.

We observed that early during learning, dopamine novelty responses were large and they slowly decreased over consecutive trials, due to a decrease in stimulus novelty as suggested previously (*Horvitz et al., 1997*; *Schultz, 1998*; *Kakade and Dayan, 2002*; *Gunaydin et al., 2014*). In both tasks, the novelty signals were mainly present in initial component of neuronal responses to cues. We used RL models to investigate how these novelty signals affected the neural and behavioural computation of value. In principle, novelty can be incorporated into RL models in two ways: (1) novelty directly augments the value function, thus increasing the predicted value and distorting future value and prediction error computations, or (2) novelty promotes exploration (in a choice setting) but does not distort value and prediction error computation (*Kakade and Dayan, 2002*). If novelty increased value estimates early in the learning session (i.e. an optimistic value initialisation), then positive prediction errors at the reward time should be very small in early trials and should slowly grow over trials, as optimism faded. Similarly, negative prediction errors would appear as strong suppressions which would be mitigated later. However, our results showed the opposite. We observed a clear development of reward prediction errors depending on the learned value of cues (*Figure 3*). On the other hand, when dopamine novelty responses were large, i.e. during early trials of choice blocks, monkeys had a strong behavioural tendency to explore the unknown option (*Figure 5*). Thus, it appears that novelty increased dopamine responses to cues and was correlated with high levels of exploration, consistent with previous studies (*Costa et al., 2014*), but the neural responses did not reflect optimistic value initiation. Given the substantial projections of dopamine neurons to cortical and subcortical structures involved in decision making (*Lynd-Balta and Haber, 1994*; *Williams, 1998*), dopamine responses to novel situations might set downstream neuronal dynamics to an activity regime that is optimal for learning (*Puig and Miller, 2012*).

Previous learning studies have shown that dopamine neurons are activated by unpredictable rewards, but not by completely predicted rewards (*Mirenowicz and Schultz, 1994*). Accordingly, dopamine neurons respond most strongly to rewards delivered near the start of learning, when rewards are most unpredictable and induce positive prediction errors (*Hollerman and Schultz, 1998*). Reward responses steadily decrease as the rewards become progressively more predictable (*Hollerman and Schultz, 1998*). However, in that study a small fraction of neurons (12%) responded to fully predicted rewards. Similarly, in studies using rodent models some dopamine responses to fully predicted rewards have remained (*Pan et al., 2005*; *Cohen et al., 2012*; *Hamid et al., 2016*). Several possible mechanisms can explain dopamine responses to 'completely predicted' rewards. With regard to the two cited learning studies in primates, the former task (in which dopamine neurons did not respond to fully-predicted rewards) was a simple instrumental task (*Mirenowicz and Schultz, 1994*), whereas in the latter task the monkeys had to make a choice before performing the instrumental response (*Hollerman and Schultz, 1998*). It is therefore possible that the more

complex task context led to less subjective certainty about upcoming reward. In our study cues predicted the reward only probabilistically, not allowing us to study dopamine responses to fully predicted rewards. Nevertheless, both excitation and suppression of dopamine responses to rewards developed over trials, in a manner consistent with prediction error signalling.

Dopamine neurons respond to prediction errors elicited by conditioned stimuli, which predict the future delivery of reward (*Schultz et al., 1997*). The dopamine response to the simultaneous onset of choice options is a special case of this responding, because future reward delivery is contingent upon the choice as well as the values that are currently on offer. Previous studies of dopamine activity during choice have shown chosen value coding by dopamine signals (*Morris et al., 2006*; *Saddoris et al., 2015*), but other studies have shown coding of the best available option, irrespective of choice (*Roesch et al., 2007*). Our results confirm the chosen value character of this response and indicate that choice-dependent dopamine signals arose very early with respect to both the onset of learning block as well as the onset of choice within each trial (*Figure 7*). However, distinct from previous reports, our results indicate that the dopamine signal takes the value of both chosen and unchosen options into account, thus reflecting relative chosen value. The relative value coding nature implies that choosing the exact same option is associated with very different responses in dopamine neurons depending on the value of the alternative option. From this standpoint, our results are fundamentally compatible with a recent report (*Kishida et al., 2016*) indicating that striatal dopamine concentration in human subjects reflects standard reward prediction error as well as counterfactual prediction error (the difference between the actual outcome and outcome of the action not taken). Our findings provide a cellular correlate for this phenomenon and indicate that flexible encoding of both choice options already occurs at the level of dopamine action potentials.

Dopamine prediction error responses are well-known teaching signals. These signals are transmitted to the striatum and cortex where they would be capable to update stimulus and action values (*Reynolds et al., 2001*; *Shen et al., 2008*). Dopamine signals induce value learning (*Steinberg et al., 2013*) and are implicated in multiple aspects of goal-directed behaviour (*Schultz, 1998*; *Bromberg-Martin et al., 2010*; *Stauffer et al., 2016*). The results demonstrated in this study advance our knowledge of dopamine function by suggesting that dopamine signals might play a critical role in computing flexible values needed for economic decision making (*Padoa-Schioppa, 2011*). The fast and flexible dopamine responses we observed during choice correspond well to recent findings demonstrating the encoding of economic utility by dopamine neurons (*Lak et al., 2014*; *Stauffer et al., 2014*) and the necessity of phasic dopamine responses for consistent choices (*Zweifel et al., 2009*). Taken together, these data point to a possible function for dopamine neurons in influencing decisions, in form of updating neuronal decision mechanisms in a rapid and flexible manner.

# Materials and methods

## Animals, surgery and setup

Two male rhesus monkeys (Macaca mulatta) were used for all experiments (13.4 and 13.1 kg). All experimental protocols and procedures were approved by the Home Office of the United Kingdom. A titanium head holder (Gray Matter Research) and stainless steel recording chamber (Crist Instruments and custom made) were aseptically implanted under general anaesthesia before the experiment. The recording chamber for vertical electrode entry was centered 8 mm anterior to the interaural line. During experiments, animals sat in a primate chair (Crist Instruments) positioned 30 cm from a computer monitor. During behavioural training, testing and neuronal recording, eye position was monitored noninvasively using infrared eye tracking (ETL200; ISCAN). Licking was monitored with an infrared optical sensor positioned in front of the juice spout (V6AP; STM Sensors). Eye, lick and digital task event signals were sampled at 2 kHz. The behavioural tasks were controlled using Matlab (Mathworks Inc.) running on a Microsoft Windows XP computer.

## Behavioural tasks
### Pavlovian learning task
In each block of the experiment, the three examined probabilities (0.25, 0.50 and 0.75) of getting the 0.4 ml juice (and 0.1 ml juice otherwise) were randomly assigned to three novel cues (i.e. fractal

images). In each trial, one of the three cues was randomly chosen and was presented to the animal. The reward was delivered 2 s after the cue onset. An experimental block was typically lasted 90–120 trials, constituting 30–40 trials for each cue. Trials were separated with inter-trial intervals of 2–5 s. The learning set in this experiment included 210 novel cues, presented in 70 blocks.

## Choice learning task

The monkeys were offered a choice between a familiar cue, whose probability was known through extensive previous training (>5000 Pavlovian trials), and a novel cue, whose probability had to be learned. The familiar cue predicted 50% chance of getting 0.4 ml and 50% chance of receiving 0.1 ml. The novel cues were associated with probabilities of 0.25, 0.50 or 0.75 of receiving the large (0.4 ml) reward and 0.1 ml otherwise. In each choice trial, after successful central fixation for 0.5 s, choice options appeared on the monitor and the animal indicated its choice by a saccade towards one of the cues. The animal was allowed to saccade as soon as it wanted. The animal had to keep its gaze on the chosen cue for 0.5 s to confirm its choice. Reward was delivered 1.5 s after the choice confirmation. Trials were separated with inter-trial interval of 2–5 s. Failure to maintain the central fixation or early break of the fixation on the chosen option resulted in 6 s time-out. At the end of each block (typically 50 trials), the novel cue was replaced with another novel cue. The probabilities assigned to novel cues were randomly chosen from the three possible probabilities. The learning set in this experiment included 120 novel cues each presented against the familiar cue in a learning block.

## Reinforcement learning models

### Reinforcement learning models in the pavlovian task

We constructed standard reinforcement learning (RL) models and fitted each model onto trial-by-trial dopamine responses in order to understand which model variant could best account for development of dopamine responses during learning.

On each trial, $t$, after experiencing a cue $x_t$ and receiving an outcome, $r_t$ (0.1 or 0.4 ml), the model computes a prediction error, $\delta_t$, by comparing the $r_t$ and the value of the stimulus, $V_t(x_t)$, according to:

$$\delta_t = r_t - V_t(x_t) \tag{1}$$

The model uses the $\delta_t$ to update the value of the stimulus, as following:

$$V_{t+1}(x_t) = V_t(x_t) + \alpha\delta_t \tag{2}$$

where $\alpha$ is the learning rate. We considered three distinct forms of learning rate: (1) a fixed learning rate (i.e. constant over trials); (2) a learning rate that decayed over trials, $\alpha_t = 1/t^k$, where $k$ is a decay constant; (3) and a learning rate that was adaptively adjusted according to:

$$\alpha_{t+1}(x_t) = \eta|\delta_t| + (1 - \eta)\alpha_t(x_t) \tag{3}$$

where $\eta$ is free parameter (a constant) which defines the degree to which the learning rate used in the current trial, $\alpha_t$ should be modified based on the experienced prediction error, $\delta_t$ (**Pearce and Hall, 1980**; **Pearce et al., 1981**; **Le Pelley, 2004**). In this model variant, we allowed $\alpha_1$ (learning rate on the first trial) as a free parameter.

In order to account for the gradual decrease in dopamine responses to cues, we considered two model variants, i.e. with or without a novelty term. In principle, novelty can be incorporated into RL models in two ways: (1) novelty directly augments the value function, thus increases the predicted value and distorts future value and prediction error computations, or (2) novelty promotes exploration (in a choice setting) but does not distort value and prediction error computation (**Kakade and Dayan, 2002**). If novelty increased value estimates early in the learning session (i.e. an optimistic value initialisation), then positive prediction errors at the reward time should be very small in early trials and slowly grows over trials, as optimism faded. Similarly, negative prediction errors would appear as strong suppressions which will be mitigated later. However, our results showed the opposite. We observed a clear development of reward prediction errors depending on the learned value of cues (**Figure 3**). Accordingly, we incorporated the decaying novelty term into RL models in a way

that it does not distort value and prediction error computation. The novelty term decayed over trials according to:

$$Novelty_t = e^{((-t/\tau)/\tau)} \tag{4}$$

where $\tau$ is the decay time constant (*Kakade and Dayan, 2002*). We considered that in model variants that included novelty term, in each trial the dopamine response to cue $x_t$ (measured 0.1–0.6 s after the cue onset and thus including both novelty and value signals) reflects the sum of this novelty term (*Equation 4*) and $V_t(x_t)$. However, in model variants without the novelty term, dopamine response to cue only reflects $V_t(x_t)$ Note that in models with the novelty term, prediction errors computation and value updating follow *Equations 1 and 2* and thus novelty term does not influence these computations. We initialised the value of all cues in a session as 0.25 ml, i.e. average value of all reward predicting cues (allowing the model to freely initialise values resulted in comparable results). In order to compare the temporal dynamics underlying novelty and values responses, we rearranged the model with fixed learning rate that included novelty term so that both its novelty and value components follow error-driven learning (fixed learning rates: $1 - e^{(-1/\tau_{early}^2)}$ and $1 - e^{(-1/\tau_{late}^2)}$, $V_0 = 1$ and 0.25 and $r_t = 0$ and actual delivered rewards, respectively). We fit this model to trial-by-trial neuronal responses (see below) and compared the recovered learning rates, reasoning that they should be the same if both response components follow similar temporal dynamics.

In order to fit the models directly onto dopamine responses, we used the trial-by-trial rewards actually delivered to the animal during each session as the input to each model, and then optimized the free parameters of each model (*Supplementary file 1*) to minimize the difference between dopamine responses to cues (measured 0.1–0.6 s after the cue onset, thus including both novelty and value component) and model's estimates of novelty + value (or only value estimates in models without the novelty term). We estimated these parameters for each learning session using a maximum likelihood procedure. To do so, we used an unconstrained Nelder–Mead search algorithm (MATLAB: fminsearch). To compare the fitting of different models on dopamine responses, we used Bayesian Information criterion (BIC); lower BIC values indicates a better fit of model on the data. Following this fitting, we regressed early neuronal responses measured 0.1–0.2 s after cue onset (i.e. dopamine novelty responses) onto novelty estimates from the superior model (*Figure 4—figure supplement 1B*). We regressed neuronal responses measured 0.2–0.6 s after the cue onset onto value estimates of the superior model. In order to have models estimates and dopamine activity on the same scale (for illustrations in *Figure 4B* and *Figure 4—figure supplement 1A* and regression analyses), we added the average normalized firing rate of neuron recorded in a session (i.e. a constant) to the model's value estimates of that session.

## Reinforcement learning models in the choice task

We constructed standard reinforcement learning (RL) models to examine animals' choices during learning and to acquire trial-by-trial estimate of chosen and unchosen values (*Figure 8—figure supplement 1*). Similar to models fitted on neuronal responses in the Pavlovian task, we constructed six variations of RL model differing in their learning rates (fixed, decaying over trial or adaptive) and inclusion of a novelty term.

The models comprised two value functions ($V_t(f)$ and $V_t(n)$) representing the learned values of the familiar and novel cues on trial $t$, respectively. In each trial, the probability that the model chooses the novel cue over the familiar cue was estimated by the softmax rule (*Sutton and Barto, 1998*) as follows:

$$p_t = \frac{e^{V_t(n)/\beta}}{e^{V_t(n)/\beta} + e^{V_t(f)/\beta}} \tag{5}$$

where $\beta$, the temperature parameter of the softmax rule, determines the level of choice randomness. Note that in models that included a novelty term, the softmax operation was performed over $V_t(f)$ and $V_t(n) + Novelty_t$, where $Novelty_t$ are computed according to *Equation 4*. This arrangement promotes choices of novel cues in initial trials of learning without influencing value and prediction error computations.

In each trial, upon making a choice and receiving an outcome, the value of the chosen option on that trial, $V_t(chosen_t)$, was updated according the reward prediction error, as follows:

$$V_{t+1}(chosen_t) = V_t(chosen_t) + \alpha[r_t - V_t(chosen_t)] \tag{6}$$

where $r_t$ indicates the size of reward received in trial $t$ (0.1 and 0.4 ml), $\alpha$ denotes the learning rate (fixed, decaying or adaptive in different model variants) and the prediction error, $\delta_t = [r_t - V_t(chosen_t)]$, indicates the difference between the expected and realized reward sizes. Given that in our experiments we had familiar (over-trained) and novel cues, it is conceivable that animals updated the value of these two cues with different rates. Thus, we allowed each of the model variants to have two different learning rates (one for each cue) in each learning block (see *Supplementary file 2*).

We estimated the free parameters of each model variants (*Supplementary file 2*) for each learning session using a maximum likelihood procedure. To do this, we used an unconstrained Nelder–Mead search algorithm (MATLAB: fminsearch). To compare the fitting of different models on behavioural choices, we used Bayesian Information criterion (BIC). Similar to models used for Pavlovian data, in model variants that included an adaptive learning rate (Equation 3), we allowed $\alpha_1$ (learning rate on the first trial) as a free parameter. We initialised the value of familiar and novel cues in a session as 0.25 ml (allowing the model to freely initialise values resulted in comparable results). Following this fitting, we regressed neuronal responses measured 0.4–0.65 s after the cue (i.e. dopamine value responses) onto chosen, unchosen and relative chosen value estimated from the superior model (*Figure 8*). We also regressed neuronal responses measured 0.1–0.2 s after cue onset (i.e. dopamine novelty responses) onto novelty estimates of the superior model (*Figure 8—figure supplement 2B*).

## Neuronal data acquisition and analysis of neuronal data

Custom-made, movable, glass-insulated, platinum-plated tungsten microelectrodes were positioned inside a stainless steel guide cannula and advanced by an oil-driven micromanipulator (Narishige). Action potentials from single neurons were amplified, filtered (band-pass 100 Hz to 3 kHz), and converted into digital pulses when passing an adjustable time–amplitude threshold (Bak Electronics). We stored both analog and digitized data on a computer using Matlab (Mathworks Inc.).

Dopamine neurons were functionally localized with respect to (a) the trigeminal somatosensory thalamus explored in awake animals and under general anaesthesia (very small perioral and intraoral receptive fields, high proportion of tonic responses, 2–3 mm dorsoventral extent), (b) tonically position coding ocular motor neurons and (c) phasically direction coding ocular premotor neurons in awake animals. Individual dopamine neurons were identified using established criteria of long waveform (>2.5 ms) and low baseline firing (<8 impulses/s) (*Schultz and Romo, 1987*). Following the standard sample size used in studies investigating neuronal responses in non-human primates, we recorded extracellular activity from 169 dopamine neurons in two monkeys (Pavlovian task: 38 and 32 neurons in monkey A and B; Choice task: 57 and 42 neurons in monkey A and B, respectively). Most neurons that met these criteria showed the typical phasic activation after unexpected reward, which we used as a fourth criterion for inclusion in data analysis.

We constructed Peri-stimulus time histograms (PSTHs) by aligning the neuronal impulses to task events and then averaging across multiple trials. The impulse rates were calculated in non-overlapping time bins of 10 ms. PSTHs were smoothed using a moving average of 70 ms for display purposes. The analysis of neuronal data used defined time windows that included the major positive and negative response components following cue onset and juice delivery, as detailed for each analysis and each figure caption.

To quantify the development of probability-dependent dopamine responses over trials, we employed one-way ANOVA, which we serially applied to trial-by-trial population responses, i.e., to responses of all neurons in trial 1, trial 2, etc. Likewise, for quantification of the time course that dopamine responses differentiate in relation to animal's choice, we used a Mann-Whitney U test on the neuronal population responses (10 ms non-overlapping window of analysis starting 200 ms before the choice). In order to quantify the differences among responses to cues for each cell recorded in the Pavlovian task, we used a one-way ANOVA on neuronal responses from sixth to last trial of each session. In order to quantify the changes of dopamine novelty responses over trials we fitted a power function ($t^n$, where t represents trial number) on normalized neuronal responses of each cell. For this fitting, responses of each neuron were normalized to its response on the first trials

of the learning block. This fit results in negative or positive values of $n$ for neurons that exhibit a decreasing or increasing cue-evoked response over trials, respectively. We used 95% confidence interval of the fit to acquire statistical significance. In order to test whether dopamine novelty responses (in the Pavlovian task) better reflect cue repetition or progress through the block (i.e. trial number), we regressed neuronal responses on number of times the cues were seen and also on the trial number in the block (for this analysis we only focused on first 10 trials of the block to better dissociate these two variables). To examine the contribution of early and late components of dopamine cue responses to prediction error computation at reward time, we employed a multiple linear regression analysis. In order to relate neuronal response in the Pavlovian task to RL model estimates, we used single linear regressions (see Reinforcement learning models section). To relate neuronal response in the choice task to RL model fits on the behavioural choice data, we used single linear regression analysis both on neuronal population response as well as on responses of each dopamine neuron (see Reinforcement learning models section).

### Normalization of neuronal responses

In order to quantify novelty response decay of each cell throughout learning using the fitting described above (*Figure 2B*), responses of each neuron was normalized to its response on the first trials of the learning block. In all other analyses, we divided spike count of each neuron in the analysis window to the spike count of the same neuron in the control window that immediately preceded each respective task event and had identical duration. Thus, a neuronal response that was not modulated by a task event had a normalized activity equal to one for that task event.

## Acknowledgements

This work was supported by Sir Henry Wellcome Trust Postdoctoral Fellowship to AL, and by grants from the Wellcome Trust, European Research Council, and National Institutes of Health Caltech Conte Center to WS.

## Additional information

### Competing interests
WS: Reviewing editor, *eLife*. The other authors declare that no competing interests exist.

### Funding

| Funder | Grant reference number | Author |
| --- | --- | --- |
| Wellcome | WT106101 | Armin Lak |
| Wellcome | | Wolfram Schultz |
| European Research Council | | Wolfram Schultz |

The funders had no role in study design, data collection and interpretation, or the decision to submit the work for publication.

### Author contributions
AL, WRS, Conception and design, Acquisition of data, Analysis and interpretation of data, Drafting or revising the article; WS, Conception and design, Analysis and interpretation of data, Drafting or revising the article

### Author ORCIDs
Armin Lak, http://orcid.org/0000-0003-1926-5458
Wolfram Schultz, http://orcid.org/0000-0002-8530-4518

### Ethics
Animal experimentation: All experimental protocols and procedures were approved by the Home Office of the United Kingdom (project licence number: 80 / 2416 and 70 /8295).

## Additional files

**Supplementary files**

• Supplementary file 1. Estimated parameters for six RL models fitted on dopamine responses in the Pavlovian task.

• Supplementary file 2. Estimated parameters for six RL models fitted on monkeys' choices.

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
