## [Decision Letter]

Thank you for submitting your article "Dopamine neurons learn relative chosen value from probabilistic rewards" for consideration by *eLife*. Your article has been reviewed by two peer reviewers, including Bruno Averbeck (Reviewer #1), and the evaluation has been overseen by a Reviewing Editor and David Van Essen as the Senior Editor.

The reviewers have discussed the reviews with one another and the Reviewing Editor has drafted this decision to help you prepare a revised submission.

Summary:

The authors describe data from a study in which they examined dopamine neuron responses and behavior while monkeys learned probabilistic reward associations of cues. They found that in both Pavlovian and choice tasks, that animals acquired behavioral responses consistent with the reward rates associated with each cue. In addition, dopamine neuron responses at the time of cue presentation and at the time of reward delivery reflected the reward probability associated with the stimulus. Interestingly, in the choice task, the early dopamine response reflected specifically cue novelty, and a later component reflected cue value. In addition, the dopamine neuron responses that reflected cue value most specifically reflected the difference between chosen and unchosen value. Most of the published data on the responses of dopamine neurons comes from tasks where the cue-outcome contingencies are highly overlearned, and there is even less data in choice tasks. The current manuscript fills that gap and finds results consistent with the reward-prediction error hypothesis.

Essential revisions:

1) The reviewers and the editor agreed that the findings are of interest to the community studying dopaminergic involvement in reward learning and choice, but we all also had reservations about the main novel contribution here, given that the main component parts of the study (dopamine responses to novelty and to chosen value) had been described in separate studies previously by this group and others. The added contribution of showing these two together during de novo reward learning does not by itself constitute an *eLife* paper. The main new result seems to be that dopamine responses to cues in the choice task reflect the difference between chosen and unchosen values, rather than chosen values alone as had been previously reported, but this in itself is also somewhat incremental. (This finding does not actually depend on the learning aspect of the design at all, and should be present but unnoticed in previous datasets with overtrained options.)

On the other hand, the article provides a somewhat clearer look at the novelty responses and how they trade off over time with learned value, compared with what had been previously available. A revised article should leverage this potential to greater effect in a way that could inform the literature so that the whole is greater than the sum of its parts. It would be useful to deliver on the novel aspects of this study by examining the trial-trial development of the dopamine responses more carefully in both experiments by fitting RL models to the neural responses directly, not just the choices, and by performing a regression of lagged outcomes onto the cue responses, somewhat in the manner of Bayer and Glimcher (2005). This should shed a lot of light on aspects of the model like the learning rate issues - how does sensitivity to previous outcomes decay with delay? Is this a Pearce-Hall like error sensitive rule, or more like a decaying learning rate? - and how the novelty responses trade off with value (can this be understood as optimistic value initialization decaying with the same learning rate by which the values come online, or is it a separate process with a distinct time constant?). The data point particularly to interesting and novel answers to the latter, but these aren't yet exposed as clearly as they could be.

2) The Introduction is reasonably complete, although there is a voltammetry paper that looks at probabilistic response learning, Hamid et al., Nature Neurosci, 2016. It's not visual cues it's leverside, but it certainly looks at learning. In addition there is a dopamine pharmacology paper by Costa et al., Beh Neurosci, 2014 that looks at the effects of increasing dopamine levels by blocking reuptake on novelty preference.

3) It's also interesting that, for example, Hamid et al. (and other rodent voltammetry papers) find persistent responses even to fully predicted rewards, whereas reward predictions in the current manuscript appear to be fully modulated by predictions. Some brief speculation on this in the discussion might be useful. From the current manuscript it's apparently not due to overtraining in monkeys compared to rodents, which one might think underlies these differences. But here, in Figure 2 and Figure 4 it looks as though there are standard RPE responses at time of reward even with within session learning. Although this study doesn't have 100% predictive cues (and hence we can't tell what would happen in that case), perhaps the authors could fit a line to the responses in 4D/Large and try to guess the intercept. Does extrapolated response imply zero error for 100% predicted reward? Are there inhibitory responses for small rewards? In both experiments? (However, the fact that the responses to 4D/small are mostly above 1 - which I think is baseline - implies that even for smaller-than-expected rewards the response is excitatory which is actually not consistent with the PE story and possibly more like rodents?) Further, the example shown is for the large drop but they don't show us a response for the small drop. And those are just examples. It wasn't clear how the responses were normalized for the plots showing responses to rewards across trials.

[Editors' note: further revisions were requested prior to acceptance, as described below.]

Thank you for resubmitting your work entitled "Dopamine neurons learn relative chosen value from probabilistic rewards" for further consideration at *eLife*. Your revised article has been favorably evaluated by David Van Essen as the Senior editor, Reviewing editor Michael Frank, and two reviewers.

Both Reviewers and the editor agree that the manuscript has been much improved. There are some remaining issues that need to be addressed before acceptance, as outlined below:

Both reviewers request additional discussion that could sharpen your presentation of the data

Reviewer 2 points out that one can recast the decaying novelty effect in terms of a learning model to determine whether the effective learning rate is consistent with optimistic initialization model. They did this in the review for the Rescorla Wagner model (and you could similarly acknowledge this equivalency and the resulting interpretation of the learning rates).

However, they also point out that given your analysis favors the Pearce-Hall model with adaptive learning rates, it would be even better to run the model where the novelty bonus is decayed by the PH learning rate, and see how that matches the neural novelty response time-course. This analysis strikes me as potentially useful, but you should take this as a suggestion rather than essential revision.

Reviewer #1:

The authors have provided detailed replies to my comments. The article provides a nice advance on our knowledge of dopamine neurons and a much more detailed examination of actual responses during learning than was previously available. I have only a few remaining comments.

1) Given the late responses of the dopamine neurons to cue value (400-650 ms), do the authors think these signals are actually related to choices? Or are they a prediction of outcome? The reaction times are slow (and these are much slower than what is frequently seen in decision making tasks, despite the cited example) so they could be. Some brief speculation on this might be useful. It is mentioned near the end of the discussion. Coupled with the late value related responses to the cues is that fact that dopamine acts through second messenger systems (assuming a co-released neuro-transmitter is not mediating these effects) and the second messenger systems add another 100 ms or more to the responses of the system to dopamine. Given Figure 7 it seems unlikely that dopamine can have much effect on the actual choices.

2) Why do the RPE responses develop more slowly than the cue value responses, across trials? In other words, cue value responses develop in the first few trials (Figure 6), whereas RPE consistent signals do not develop until late in the blocks (Figure 6).

Reviewer #2:

I think this article is much improved; the authors have fulfilled all the requests of the reviewers and editors. The article now gives a clear and full description of the dopamine response during initial formation of probabilistic associations. There are a number of interesting results as part of this; the one that now comes across as the clearest and most sustained is further clarification of the distinction between early and late phase dopamine responses.

I have a couple of small suggestions for final revision.

1) I apologize that my suggestion about comparing the novelty response to the learning rate was not clear. To my mind, it is not all that meaningful to compare the time dynamics of the learning rate to the time dynamics of novelty response (Figure 4) as these are apples and oranges.

What I meant was that, if the declining novelty response arose from error-driven learning (as it would if it were a component of optimistically initialized values), then its estimated decay constant implies an equivalent learning rate. And comparing *this* time constant to the learning rate for value acquisition might provide additional evidence for dissociating this dynamics from the value learning dynamics.

In particular, the exponentially decaying novelty response model, e(−t/τ2), can equivalently be written as a PE-driven learning model, with *V(0)* = 1, fixed learning rate α=1−e(−1/τ2) and *r(t)* = 0 for all *t*. Because of the linearity of the RW model, this also means that if you added one to initial values *V_0_* (optimistic initialization) and learned from some actual timeseries of rewards with that same fixed learning rate, the resulting value timeseries would equal the original value timeseries (without optimistic initialization), plus the same exponentially decaying novelty bonus.

In other words the optimistic initialization model implies that the novelty bonus contribution should decay with a time constant given by the learning rate; if these two time constants are far from one another, this would be evidence against that model.

If my math is correct, the novelty "learning rate" is consistently faster than the experimental learning rate: perhaps reasonably close for Experiment 2 but far for Experiment 1. For this I used the RW&N model since it assumes constant and therefore comparable learning rates across both processes. However using the preferred PH&N model, one could also model the effect of the PH-adjusting learning rate on a decaying optimistic novelty bonus (rather than constant exponential decay) and see how this matched the actual novelty response.

Anyway this is the sort of comparison I thought would be useful to work through.

2) The point that optimistic value initialization also implies a mirror-image negative contribution to the reward-time prediction error is an important one, and again speaks to the dissociation of the novelty response from the value learning. However, I think the question whether this is really present in the data is a little subtler than the discussion makes it out to be. Figure 3 actually does seem to show, for the worst cue, positive errors growing gradually and negative errors diminishing gradually. I think this figure implies that *V(0)* is initialized somewhere near the value of the middle cue. The question is whether this matches the dynamics of learning implied by the late phase component, Figure 2 alone, or whether it better matches what would be predicted from those values plus an additional negative contribution from the early-phase value component 2B. I think Figure 2 also implies values initialized near the value of the middle cue, thus no additional contribution from the novelty response is needed to explain Figure 3, supporting the authors interpretation.

---

## [Author Response]

*[….] Essential revisions:*

*1) The reviewers and the editor agreed that the findings are of interest to the community studying dopaminergic involvement in reward learning and choice, but we all also had reservations about the main novel contribution here, given that the main component parts of the study (dopamine responses to novelty and to chosen value) had been described in separate studies previously by this group and others. The added contribution of showing these two together during* de novo *reward learning does not by itself constitute an eLife paper. The main new result seems to be that dopamine responses to cues in the choice task reflect the difference between chosen and unchosen values, rather than chosen values alone as had been previously reported, but this in itself is also somewhat incremental. (This finding does not actually depend on the learning aspect of the design at all, and should be present but unnoticed in previous datasets with overtrained options.)*

We have extensively modified the Introduction to clearly spell out the questions that our manuscript asks and also referred to these topics one by one in the Discussion section. Additional analyses that we have performed based on reviewers suggestions (Figure 2, Figure 3, Figure 4 and Figure 4—figure supplement 1 and Figure 8—figure supplement 1 and Figure 8—figure supplement 2) together with changes to the revised Introduction and Discussion better communicate the advances our manuscript offers.

*On the other hand, the article provides a somewhat clearer look at the novelty responses and how they trade off over time with learned value, compared with what had been previously available. A revised article should leverage this potential to greater effect in a way that could inform the literature so that the whole is greater than the sum of its parts. It would be useful to deliver on the novel aspects of this study by examining the trial-trial development of the dopamine responses more carefully in both experiments by fitting RL models to the neural responses directly, not just the choices, and by performing a regression of lagged outcomes onto the cue responses, somewhat in the manner of Bayer and Glimcher (2005). This should shed a lot of light on aspects of the model like the learning rate issues -how does sensitivity to previous outcomes decay with delay? Is this a Pearce-Hall like error sensitive rule, or more like a decaying learning rate? - and how the novelty responses trade off with value (can this be understood as optimistic value initialization decaying with the same learning rate by which the values come online, or is it a separate process with a distinct time constant?). The data point particularly to interesting and novel answers to the latter, but these aren't yet exposed as clearly as they could be.*

Here we list additional analyses/modelling that we have performed to accommodate reviewers’ advices.

Fitting RL model directly on neuronal data: we have fitted six variants of RL models directly to the neuronal responses recorded in the Pavlovian experiment. These models have three different possible learning rates (Fixed, decaying over trials and Pearce-Hall) and each includes a novelty term or not, resulting in six model variants. These models and their fits on dopamine responses are extensively described in Figure 4, Results section, Materials and methods section and [Supplementary-material SD1-data]. Note that in our revised manuscript, we are using similar RL model variants for modelling animals’ choice behaviour (Figure 8—figure supplement 1 and Figure 8—figure supplement 2, [Supplementary-material SD2-data], Materials and methods section).

The learning rate:as reviewers suggested, in both modelling sections (in Pavlovian and choice tasks), we now use and compare three different learning rates: Fixed, Decaying over trials (1/n^k^) and Pearce-Hall (Figure 4, Figure 4—figure supplement 1, Figure 8—figure supplement 1 and Figure 8—figure supplement 2, [Supplementary-material SD1-data] and [Supplementary-material SD2-data]). This has been clarified in the main text.

Dissociating novelty and value signals: to examine the trade off between value and novelty signals, we have added a new analysis of the Pavlovian data in which we broke dopamine responses to early and late components (Figure 2), analogous to Figure 6 of the choice task. Moreover, in both model fittings, we could show that model’s estimate of novelty can account for dopamine novelty responses and model’s estimate of value can account for dopamine value signals (Pavlovian: Figure 4 and Figure 4—figure supplement 1, Choice: Figure 8 and Figure 8—figure supplement 2). Furthermore, our modelling results showed that value updating continues after novelty signals faded, indicating two different time constants for these two parameters (See Figure 4 and Results section).

Optimistic values: In principle, novelty can be incorporated into RL models in two ways: (1) novelty directly augments the value function, thus increasing the predicted value and distorting future value and prediction error computations, or (2) novelty promotes exploration (in a choice setting) but does not distort value and prediction error computation (Kakade and Dayan, 2002). If novelty increased value estimates early in the learning session (i.e. an optimistic value initialization), then positive prediction errors at the reward time should be very small in early trials and slowly grows over trials (as optimism faded). Similarly, negative prediction errors would appear as strong suppressions which be mitigated later, as optimism faded. However, our results showed the opposite. We observed a clear development of reward prediction errors depending on the learned value of cues (Figure 3). On the other hand, when dopamine novelty signals are large, i.e. during early trials of choice blocks, we observed a strong behavioural tendency to explore the unknown option (Figure 5). Thus, it appears that while novelty increased dopamine responses to cues and was correlated with high levels of exploration, the neural value response did not reflect optimistic value initiation. Accordingly, we incorporated the decaying novelty term into RL models in a way that it does not distort value and prediction error computation (Materials and methods section). We have discussed this in our revised Discussion.

*2) The Introduction is reasonably complete, although there is a voltammetry paper that looks at probabilistic response learning, Hamid et al., Nature Neurosci, 2016. It's not visual cues it's leverside, but it certainly looks at learning. In addition there is a dopamine pharmacology paper by Costa et al., Beh Neurosci, 2014 that looks at the effects of increasing dopamine levels by blocking reuptake on novelty preference.*

We have included the citation to Hamid et al. 2016 to our revised Introduction and we have cited Costa et al. 2014 in our revised Discussion.

*3) It's also interesting that, for example, Hamid et al. (and other rodent voltammetry papers) find persistent responses even to fully predicted rewards, whereas reward predictions in the current manuscript appear to be fully modulated by predictions. Some brief speculation on this in the discussion might be useful. From the current manuscript it's apparently not due to overtraining in monkeys compared to rodents, which one might think underlies these differences. But here, in Figure 2 and Figure 4 it looks as though there are standard RPE responses at time of reward even with within session learning. Although this study doesn't have 100% predictive cues (and hence we can't tell what would happen in that case), perhaps the authors could fit a line to the responses in 4D/Large and try to guess the intercept. Does extrapolated response imply zero error for 100% predicted reward?*

Previous studies have examined learning of stimulus reward associations where the stimulus predicted the reward size and timing with certainty (Hollerman and Schultz, 1998; Mirenowicz and Schultz, 1994). In those studies, the majority of dopamine neurons stopped responding to fully predicted rewards. This response change occurred in line with behavioural learning, on the order of 20 trials. A minority of dopamine neurons (12%) maintained some responding to fully predicted rewards, but only in the choice task (Hollerman and Schultz, 1998). We have added this to the Discussion section. In that same new Discussion paragraph, we have expanded on other differences between rodent and monkey experiments that could affect learned responses.

*Are there inhibitory responses for small rewards? In both experiments? (However, the fact that the responses to 4D/small are mostly above 1 - which I think is baseline - implies that even for smaller-than-expected rewards the response is excitatory which is actually not consistent with the PE story and possibly more like rodents?) Further, the example shown is for the large drop but they don't show us a response for the small drop. And those are just examples. It wasn't clear how the responses were normalized for the plots showing responses to rewards across trials.*

Inhibitory responses to small rewards: we observed inhibitory responses to negative PE in both tasks, and we have modified our analysis to reflect this (Figure 3, Figure 6). Previous studies have shown that pauses in dopamine firing (inhibitions) to negative PE are often followed by rebound responses (Fiorillo et al., 2013). This late rebound response can easily conceal the pause regularly observed earlier in the response. Accordingly, we have revised our analysis to use differently sized analysis windows for positive and negative PE responses, which are illustrated in Figure 3 and Figure 6 and are described in the main text and in the figure captions (caption of Figure 3 and caption of Figure 6).

Response normalization: To normalize neuronal responses, we divided spikes counted during the analysis window onto spikes counted in the control window that immediately preceded the task event of interest. Thus, 1 indicates no change in the neuronal response. We have described details of time windows used for each analysis in each figure and its corresponding figure caption and we clarified the normalization process in subsection “Normalization of neuronal responses”.

[Editors' note: further revisions were requested prior to acceptance, as described below.]

*[…] Both Reviewers and the editor agree that the manuscript has been much improved. There are some remaining issues that need to be addressed before acceptance, as outlined below:*

*Both reviewers request additional discussion that could sharpen your presentation of the data*

We have included the suggested discussion points to our revised manuscript. Please see below.

*Reviewer 2 points out that one can recast the decaying novelty effect in terms of a learning model to determine whether the effective learning rate is consistent with optimistic initialization model. They did this in the review for the Rescorla Wagner model (and you could similarly acknowledge this equivalency and the resulting interpretation of the learning rates).*

*However, they also point out that given your analysis favors the Pearce-Hall model with adaptive learning rates, it would be even better to run the model where the novelty bonus is decayed by the PH learning rate, and see how that matches the neural novelty response time-course. This analysis strikes me as potentially useful, but you should take this as a suggestion rather than essential revision.*

We have performed the additional analyses that the Reviewer has suggested and included them in the revised. Please see below.

*Reviewer #1:*

*The authors have provided detailed replies to my comments. The article provides a nice advance on our knowledge of dopamine neurons and a much more detailed examination of actual responses during learning than was previously available. I have only a few remaining comments.*

*1) Given the late responses of the dopamine neurons to cue value (400-650 ms), do the authors think these signals are actually related to choices? Or are they a prediction of outcome? The reaction times are slow (and these are much slower than what is frequently seen in decision making tasks, despite the cited example) so they could be. Some brief speculation on this might be useful. It is mentioned near the end of the discussion. Coupled with the late value related responses to the cues is that fact that dopamine acts through second messenger systems (assuming a co-released neuro-transmitter is not mediating these effects) and the second messenger systems add another 100 ms or more to the responses of the system to dopamine. Given Figure 7 it seems unlikely that dopamine can have much effect on the actual choices.*

We don’t think that the observed dopamine responses could influence current choice computations. Despite the fact that the responses differentiate around 100 ms before the saccade onset, they most probably reflect prediction errors in relation to an already computed choice. We have discussed this point in our revised manuscript.

“In the learning experiment that involved choices, the neuronal responses rapidly differentiated to reflect animal’s choice. These differential responses, despite appearing more than 100 ms prior to overt behaviour, reflect prediction errors, in relation to an already computed choice, and thus might not directly participate in current choice computation.”

*2) Why do the RPE responses develop more slowly than the cue value responses, across trials? In other words, cue value responses develop in the first few trials (Figure 6), whereas RPE consistent signals do not develop until late in the blocks (Figure 6).*

This temporal mismatch most probably points to the fact that value responses to cues, at least partially, could be originated from other brain regions. We have discussed this point in the Discussion section.

“Interestingly, in both Pavlovian and choice tasks, behavioural preferences as well as neuronal responses to cues reflected reward probability earlier during learning than the neuronal reward responses. This temporal difference might suggest an origin of behavioural preferences and acquired dopamine cue responses in other brain structures, rather than relying primarily on dopamine reward prediction error signals.”

*Reviewer #2:*

*I think this article is much improved; the authors have fulfilled all the requests of the reviewers and editors. The article now gives a clear and full description of the dopamine response during initial formation of probabilistic associations. There are a number of interesting results as part of this; the one that now comes across as the clearest and most sustained is further clarification of the distinction between early and late phase dopamine responses.*

*I have a couple of small suggestions for final revision.*

*1) I apologize that my suggestion about comparing the novelty response to the learning rate was not clear. To my mind, it is not all that meaningful to compare the time dynamics of the learning rate to the time dynamics of novelty response (Figure 4) as these are apples and oranges.*

*What I meant was that, if the declining novelty response arose from error-driven learning (as it would if it were a component of optimistically initialized values), then its estimated decay constant implies an equivalent learning rate. And comparing this time constant to the learning rate for value acquisition might provide additional evidence for dissociating this dynamics from the value learning dynamics.*

In particular, the exponentially decaying novelty response model, e(−t/τ2), can equivalently be written as a PE-driven learning model, with V(0) = 1, fixed learning rate α=1−e(−1/τ2)andr(t) = 0 for all t. Because of the linearity of the RW model, this also means that if you added one to initial values V_0_ (optimistic initialization) and learned from some actual time series of rewards with that same fixed learning rate, the resulting value time series would equal the original value time series (without optimistic initialization), plus the same exponentially decaying novelty bonus.

In other words the optimistic initialization model implies that the novelty bonus contribution should decay with a time constant given by the learning rate; if these two time constants are far from one another, this would be evidence against that model.

*If my math is correct, the novelty "learning rate" is consistently faster than the experimental learning rate: perhaps reasonably close for Experiment 2 but far for Experiment 1. For this I used the RW&N model since it assumes constant and therefore comparable learning rates across both processes. However using the preferred PH&N model, one could also model the effect of the PH-adjusting learning rate on a decaying optimistic novelty bonus (rather than constant exponential decay) and see how this matched the actual novelty response.*

Anyway this is the sort of comparison I thought would be useful to work through.

We have performed the rewriting of the model as the reviewer suggested and examined the time constant of a model that includes novelty as an error-driven learning process. We compared the recovered learning rate for early and late responses, reasoning that these learning constants should be statistically equal, if these two processes were to follow the same temporal dynamics. As new Figure 4—figure supplement 1 shows, the learning rate for early responses are larger than those of late responses. We have performed this analysis for RW&N model and included the results in the text. We have removed our original comparison between temporal dynamics of novelty response and learning rate (previous Figure 4, right).

*2) The point that optimistic value initialization also implies a mirror-image negative contribution to the reward-time prediction error is an important one, and again speaks to the dissociation of the novelty response from the value learning. However, I think the question whether this is really present in the data is a little subtler than the discussion makes it out to be. Figure 3 actually does seem to show, for the worst cue, positive errors growing gradually and negative errors diminishing gradually. I think this figure implies that V(0) is initialized somewhere near the value of the middle cue. The question is whether this matches the dynamics of learning implied by the late phase component, Figure 2 alone, or whether it better matches what would be predicted from those values plus an additional negative contribution from the early-phase value component 2B. I think Figure 2 also implies values initialized near the value of the middle cue, thus no additional contribution from the novelty response is needed to explain Figure 3, supporting the authors interpretation.*

We have added an extra analysis that could further clarify our findings. The multiple linear regression analysis suggests that responses to reward are influenced by reward size and also late dopamine responses to cues, with no contribution of early cue responses.

“The development of dopamine responses to rewards further suggests that early and late responses to cues convey distinct signals. If early responses to cues contained predictive values signals (i.e. reflecting an optimistic value initialisation), such signals should have contributed to prediction error computations at reward time. However, the pattern of neuronal reward prediction errors (Figure 3) suggests that these responses were computed in relation to late responses to cues, and reflected cue values initialised around the average value of all cues. Accordingly, neuronal responses to rewards were accounted for by the late component of neuronal responses to cues as well as the received reward size, with no significant contribution from the early component of cue responses (p = 0.0001, 0.43 and 0.021 for reward size, early and late cue responses, respectively; multiple linear regression).”